# Analysis of multi-level barriers to physical activity among nursing students using regularized regression

Muge Capan[1]*, Lily Bigelow[1], Yukti Kathuria[1], Amanda Paluch[2‡], Joohyun Chung[3‡]

**1** Department of Mechanical and Industrial Engineering, College of Engineering, University of Massachusetts Amherst, Amherst, MA, United States of America, **2** Department of Kinesiology and Institute for Applied Life Sciences, University of Massachusetts Amherst, Amherst, MA, United States of America, **3** College of Nursing, University of Massachusetts Amherst, Amherst, MA, United States of America

☯ These authors contributed equally to this work.
‡ AP and JC also contributed equally to this work.
* mcapan@umass.edu

**Data Availability Statement:** De-identified data are uploaded as supplemental material.

**Funding:** University of Massachusetts Amherst -Elaine Marieb Center for Nursing and Engineering

## Abstract

Physical inactivity is a growing societal concern with significant impact on public health. Identifying barriers to engaging in physical activity (PA) is a critical step to recognize populations who disproportionately experience these barriers. Understanding barriers to PA holds significant importance within patient-facing healthcare professions like nursing. While determinants of PA have been widely studied, connecting individual and social factors to barriers to PA remains an understudied area among nurses. The objectives of this study are to categorize and model factors related to barriers to PA using the National Institute on Minority Health and Health Disparities (NIMHD) Research Framework. The study population includes nursing students at the study institution (N = 163). Methods include a scoring system to quantify the barriers to PA, and regularized regression models that predict this score. Key findings identify intrinsic motivation, social and emotional support, education, and the use of health technologies for tracking and decision-making purposes as significant predictors. Results can help identify future nursing workforce populations at risk of experiencing barriers to PA. Encouraging the development and employment of health-informatics solutions for monitoring, data sharing, and communication is critical to prevent barriers to PA before they become a powerful hindrance to engaging in PA.

## Introduction

Physical inactivity is a growing societal concern with significant impact on disability, burden of non-communicable diseases, mortality, and morbidity [1–3]. National guidelines in the U.S. recommend adults to engage in 150 minutes per week of moderate-intensity or 75 minutes per week of vigorous-intensity aerobic physical activity (PA) and resistance training twice per week [4]. Nurses have a high prevalence of non-adherence to PA recommendations [2]. Low levels of PA are particularly concerning in healthcare professions in direct contact with patients, families, and communities, such as nurses. Nurses represent one of the largest

Innovation Faculty Pilot Grant supporting the study entitled "Exploratory Study to Identify Multi-level Factors of Physical Health and Well-being in Nursing Profession" (06/2023 – 06/2024, PI: M. Capan).

**Competing interests:** The authors have declared that no competing interests exist.

healthcare professions in the U.S. with nearly 4.2 million registered nurses [5]. When nurses' health and well-being are compromised due to physical inactivity, it can negatively impact safety and quality of care, e.g., increased risk of clinical errors [6].

Understanding the factors acting as *barriers* to engage in PA in nursing population can improve nurses' PA. While determinants of PA have been widely studied, the question of how multi-level factors (e.g., individual, intrapersonal, environmental) result in disparities such that some populations experience more barriers to PA than others remains an unresolved question. Existing studies are limited due to low survey response rates from nurses in the workplace [7], and the lack of a comprehensive framework to connect the multi-level factors hindering PA.

The objectives of this study are to categorize multi-level factors associated with barriers to engage in PA according to selected domains and influence levels of NIMHD Research Framework, and analyze the impact of these factors on barriers to PA among nursing students. Specifically, the study questions (Q) explored in this paper are:

- **Q1**: How can we quantify self-reported barriers to engage in PA using a scoring system?

- **Q2**: How can we use individual, interpersonal, behavioral and socio-cultural factors to predict this score in nursing student population?

Below we review relevant studies and their findings regarding: i) barriers to PA in nursing, ii) connecting health disparities and barriers to PA in nursing, and iii) a conceptual framework to categorize and quantify barriers to PA.

## Barriers to physical activity in nursing

Nurses self-report levels of moderate- to vigorous-intensity PA below the recommended levels for adults [8–11]. In one study, 60–74% of nurses were classified as being at high risk for poor health outcomes due to low PA levels [2], and 50% of nurses did not meet guidelines for PA [12]. Barriers to PA in the nursing population have been categorized in various ways, e.g., internal (e.g., attitudes reflecting negative attitudes towards PA) or external (e.g., physical or environmental impediments that prevent PA, lack of facilities) [13, 14]. Nurses have largely cited cost, lack of accessible and affordable exercise spaces, fear of inducing adverse health conditions (e.g., injury, fatigue, muscle soreness), physical appearance, lack of social support, lack of time, and stress as barriers to PA [11, 15, 16].

A systematic review identified several barriers to PA in nursing work environments [17]. Some examples include social barriers, such as lack of access to exercise facilities [18], organizational barriers such as long work hours, and individual barriers such as lack of motivation and self-efficacy [7]. Several studies highlighted occupational factors related to physical inactivity in nursing. Nursing profession is reported to be physically demanding [19], and many nurses experience job dissatisfaction, burnout [20], and mental health issues [11]. Nurses affected by these factors are put at risk for poor health behaviors including physical inactivity [11, 19].

Identifying the barriers to PA is particularly important for individuals preparing to enter the nursing workforce. Nursing students are also cited to be physically inactive [21, 22]. In a study of 264 nursing students, 68.6% of the students were not meeting the recommended amounts of PA [23]. A study conducted focus groups with Australian nursing students and identified individual (e.g., lack of motivation, lack of knowledge), environmental (e.g., time, finances, limited access to PA resources), and psychosocial (e.g., competing priorities, increased cognitive load, lack of social interaction and support) barriers to PA [22].

## Health disparities and barriers to physical activity

Health disparities represent differences in the incidence and prevalence of health conditions between groups. PA is influenced by health disparities, including but not limited to differences in socioeconomic status, race and ethnicity, and access to PA facilities. Disparities may occur due to some populations disproportionately experiencing barriers to PA. For example, cost- or distance-related factors limiting PA may impact socially vulnerable, minority, and/or rural populations more compared to other populations. In the context of barriers to PA, minority ethnic or racial groups, older individuals, women, low income individuals, and less educated individuals are less likely to engage in sufficient PA [24]. Furthermore, disparities such as rurality and lack of access to exercise facilities have been cited as environmental barriers to PA [24]. Health disparities and barriers to PA are intertwined, and as such, it is necessary to consider both when identifying multi-level factors that influence barriers to PA.

## The National Institute on Minority Health and Health Disparities (NIMHD) Research Framework

The National Institute on Minority Health and Health Disparities (NIMHD) framework is a conceptualization of factors impacting minority health and health disparities [25]. The framework identifies five *domains of influence* (i.e., biological, behavioral, physical/built environment, socio-cultural, and healthcare system) and four *levels of influence* (i.e., individual, interpersonal, community, and societal) (Fig 1).

Previous research has utilized various psychosocial models to explain PA engagement, including the Socioecological model, the Theory of Planned Behavior, and the Health Belief

| | | **Levels of Influence\*** | | | |
|---|---|---|---|---|---|
| | | **Individual** | **Interpersonal** | **Community** | **Societal** |
| **Domains of Influence** *(Over the Lifecourse)* | **Biological** | Biological Vulnerability and Mechanisms | Caregiver–Child Interaction Family Microbiome | Community Illness Exposure Herd Immunity | Sanitation Immunization Pathogen Exposure |
| | **Behavioral** | Health Behaviors Coping Strategies | Family Functioning School/Work Functioning | Community Functioning | Policies and Laws |
| | **Physical/Built Environment** | Personal Environment | Household Environment School/Work Environment | Community Environment Community Resources | Societal Structure |
| | **Sociocultural Environment** | Sociodemographics Limited English Cultural Identity Response to Discrimination | Social Networks Family/Peer Norms Interpersonal Discrimination | Community Norms Local Structural Discrimination | Social Norms Societal Structural Discrimination |
| | **Health Care System** | Insurance Coverage Health Literacy Treatment Preferences | Patient–Clinician Relationship Medical Decision-Making | Availability of Services Safety Net Services | Quality of Care Health Care Policies |
| **Health Outcomes** | | Individual Health | Family/ Organizational Health | Community Health | Population Health |

**Fig 1. National Institute on Minority Health and Health Disparities (2017) NIMHD Research Framework.** Retrieved from https://nimhd.nih.gov/researchFramework and reproduced with permission from the National Institute on Minority and Health Disparities, National Institute of Health.

Model [26]. Yet, to the best of our knowledge, there are no studies that utilized the NIMHD framework to connect the multi-level factors associated with experiencing barriers to PA.

The remainder of the paper is organized as follows. In Materials and Methods section, we describe the data pre-processing, descriptive, statistical and predictive methods adopted in our research. Specifically, we developed a scoring system based on validated survey tools to quantify barriers to PA, and developed and validated regularized regression models that aim to predict this score. Next, we present the modeling results. Finally, in Discussion section we conclude and discuss methodological and practical contributions, limitations, and future research directions to address disparities in experiencing barriers to PA.

## Materials and methods

### Study design

This is a cross-sectional study with a set of questionnaire surveys conducted at the University of Massachusetts Amherst, Amherst MA, between September and October 2023. The study was approved by the Institutional Review Board (IRB) of University of Massachusetts Amherst. The recruitment period for this study started on September 15, 2023 and ended on October 31, 2023. Participants provided written informed consent which was documented in Research Electronic Data Capture (REDCap).

### Study population

Inclusion criterion of this study is being currently enrolled in a nursing degree program at the study institution. The nursing degree program includes the Bachelors (BS) in Nursing (4-year undergraduate program) and Accelerated BS Nursing program (18-months program for individuals who have completed another BS degree and are pursuing a nursing BS degree). Exclusion criterion is not consenting to participate.

### Survey development

Study survey was designed in REDCap using validated questionnaires on demographics, Social Determinants of Health Collection of the PhenX Toolkit, health behaviors (e.g., sleep, diet), physical activity (e.g., type, frequency, duration, and intensity), psychological health (e.g., anxiety, depression using the Patient Health Questionnaire–9), and access to and use of healthcare technology, among other factors. REDCap is a secure web application for building and managing online surveys and databases [27, 28]. The validated survey tools (Table 1) corresponding to domains and levels of influence from the NIMHD Research Framework (Fig 1) are listed below:

- International Physical Activity Questionnaire (IPAQ) [29]

- Motives for Physical Activity Measure (MPAM-R) [30]

**Table 1. Adapted National Institute on Minority Health and Health Disparities (NIMHD) Framework and corresponding survey tools.** IPAQ stands for International Physical Activity Questionnaire. MPAM-R stands for Motives for Physical Activity Measure—Revised. EBBS stands for Exercise Benefits/Barriers Scale. PASSS stands for hysical Activity and Social Support Scale. SDOH stands for Social Determinants of Health.

| Domains of Influence | Levels of Influence | |
|---|---|---|
| | **Individual** | **Interpersonal** |
| Behavioral | IPAQ, MPAM-R, EBBS | EBBS, PASSS |
| Sociocultural Environment | PhenX SDOH Toolkit | PhenX SDOH Toolkit |

- Exercise Benefits/Barriers Scale (EBBS) [31]

- PhenX Social Determinants of Health (SDOH) Toolkit [32]

- Physical Activity and Social Support Scale (PASSS) [33]

In total, there were 174 questions in multiple-choice, yes/no, and numeric entry formats. The IPAQ includes 27 questions across 4 domains of PA per week (i.e., job, transportation, housework, and leisure. Previous studies have established test-retest reliability for this measure [29, 34]. The MPAM-R is an instrument with 30 questions on a scale of 1–7 that are used to assess the reasons that motivate individuals in the pursuit of physical activities. The scale is based along 5 themes: fitness, appearance, competence/challenge, social and enjoyment and these motives are connected to different outcomes. Reliability and construct validity for the MPAM-R have been previously established [30, 35, 36]. The EBBS has two parts: the 29-item benefits scale and the 14-item barriers scale that measure the perception of individuals towards the benefits and barriers of exercise. The scale ranges from 1–4. Good validity and reliability for this measure has been previously established [31, 37]. The PhenX SDOH Toolkit is comprised of a core and additional specialized collection of questions with individual and structural measurement protocols. The core collection used in this study includes demographic data such as race, age, gender identity, annual family income, access to health services, among other SDOH factors. Protocols from the PhenX SDOH Toolkit are well-established and have demonstrated reliability, reproducibility, and validity [38, 39]. The PASSS consists of 20 items across 5 domains (i.e., emotional support, validation support, informational support, companionship support, instrumental support) and a scale of 1–7 that examines the perception of various support mechanisms related to engaging in PA. The PASSS has been shown to have criterion validity and internal consistency [33, 40].

Each questionnaire described above was matched to the intersection of the two selected domains (behavioral and sociocultural environment) and two selected levels of influence (individual and interpersonal) within the NIMHD framework as shown in Table 1. This matching is a critical step in predictive model development for the prediction of the outcome of interest in our study, which is the Physical Activity Barrier Score (PABS). We operationalized PABS by summing the responses to 14 questions from the EBBS that are designated to measure barriers to exercise. All 14 questions are statements on a Likert-scale on a scale from 1 (never true) to 7 (always true). We developed regression models for each domain-level intersection in Table 1 using participants' responses to the corresponding questionnaires as independent variables to predict PABS. The model development and evaluation will be further discussed in Methods section. The final version of the REDCap survey is provided in Supplementary Materials (S1 File).

### Data collection

Data was collected in four different in-person sessions during September—October 2023. Each session was coordinated and conducted by at least two of the study team members. During each data collection session, participants were informed about human subjects research and IRB. The participants were provided the consent form. Access to the REDCap survey designed for computer, tablet, and phone. Participation took on average 25–35 minutes to complete. To begin the survey, participants clicked on the survey link and/or QR code shown on the screen. They were able to save and continue where they left off if needed. All participants were asked to complete the survey by October 31, 2023. All participants who completed the survey received a $20 gift card for their participation.

## Data preprocessing

Informed consent was obtained from all study participants. R statistical software (v4.3.1) was used for data cleaning and preprocessing [41]. One participants observations were excluded for being above possible PA levels achievable in a week [29]. Approximately 14.83% of all data contained missing values, which were accounted for using various imputation methods. Median imputation was used for variables that contained values from a Likert Scale (i.e., variables from the MPAM-R, EBBS, and PASSS). Domain-specific imputation was used for the remaining numerical and categorical variables that contained a majority of missing values, including survey questions that contained branching logic. Missing values for independent variables are discussed in the Supplementary Materials (S1 Table).

Little's Test [42] was used to determine the state of missingness for our data. Missing data is classified in three ways: Missing Completely at Random (MCAR), Missing at Random (MAR), and Missing Not at Random (MNAR). MCAR is where missing data values are independent of study variables or unobserved data, MAR is missing data values are dependent on study variables, and MNAR is missing data values are dependent on study variables and unobserved data [43, 44]. Under the null hypothesis that missing data is MCAR, Little's Test calculates Chi-squared statistics based on the differences between observed and expected missingness patterns. Data is MCAR if results are non-significant, failing to reject the null hypothesis. For this study, Little's Test was performed on variables belonging to each validated tool (e.g., IPAQ, MPAM-R, EBBS, PASSS, PhenX SDOH Toolkit), using R statistical software (v.4.3.1) and package naniar (v.1.0.0) [45] using mcar_test().

Following imputation, categorical independent variables were factorized and variables containing a large number of categories were re-categorized as needed. Factorization refers to the process of grouping distinct values of a categorical variable into discrete levels, allowing qualitative data to be represented numerically. Some of the independent variables were directly derived from survey responses, others were computed using the survey responses. From the IPAQ, four variables describing domains of PA measured in metabolic equivalent (MET)-minutes/week (work, transportation, yard/domestic, and leisure) were calculated according to scoring guidelines [29]. From the MPAM-R, five variables describing types of motives (i.e., fitness, appearance, competence, social, and interest/enjoyment) were calculated by summing scores for questions related to each type of motive [30]. From the PASSS, five variables describing forms of social support (i.e., emotional, informational, instrumental, validation, and companionship) were calculated by summing scores for questions related to each social support [33].

## Descriptive and statistical analysis

In addition to questions from survey tools (Table 1), demographic factors consisting of race, age, family income, birthplace, biological sex, level of education, and home language were collected. Summary statistics and visualization of distributions of numerical variables were conducted. Non-parametric statistical tests were used to test whether there was a significant difference regarding PABS between the categories of categorical factors. Age and family income were categorized using median and quantiles of the response distributions prior to testing. The Mann Whitney U Test applies to features that only have two groups whereas the Kruskal Wallis Test is a generalization of the Mann Whitney U Test and is used to test difference with regards to PABS between levels of categorical independent variables with more than two categories.

## Predictive analysis

R statistical software (v4.3.1) was used to develop all predictive models and perform all analyses [41]. Outcome of interest was the numerical variable PABS. Predictive models were developed by first performing stepwise regression, and then implementing regularized regression with k-fold cross validation. Stepwise regression is the iterative selection of variables from a model based on the predictors' level of association with the outcome variable and statistical significance (i.e., contribution to the coefficient of determination, $R^2$, or the ability to minimize the sum of squared error) [46, 47]. There are three variable selection approaches that can be used: the iterative addition (forward selection), iterative removal (backward selection), or simultaneous addition and removal of variables (bidirectional elimination). For this study, bidirectional elimination was chosen as the selection approach and was implemented with R package MASS (v7.3.60), using the stepAIC() function [47, 48].

Regularized regression with k-fold cross validation was applied to the best fit models resulting from the stepwise regression. Regularized regression incorporates constraints to minimize the number of insignificant independent variables in a regression model, and includes Ridge, LASSO, and Elastic net regression approaches [49]. Comparatively, ordinary least squares (OLS) regression attempts to minimize the residual sums of squares (RSS) between observed and predicted values [50] and is given by the equation:

$$RSS = \sum_{i=1}^{n} \left( y_i - \beta_0 - \sum_{j=1}^{p} \beta_j x_{ij} \right)^2 \tag{1}$$

In (1), output variables are given by $y_i$, input variables by $x_{ij}$, regression coefficients $\beta_j$, and intercept $\beta_0$. Regularized regression works similarly in that it attempts to minimize RSS, yet it also applies a penalty term to the objective function that shrinks coefficients of predictors to zero. Ridge regression [51] is expressed as follows:

$$RSS + \lambda \sum_{j=1}^{p} \beta_j^2 \tag{2}$$

In (2), $\lambda \geq 0$ is the tuning parameter of the $L_2$ penalty term, which serves to control the shrinkage of $\beta_j$ towards zero, but not exactly to zero [49, 50]. LASSO regression [52] uses an $L_1$ penalty term, which can perform feature selection of variables due to its ability to shrink coefficients exactly to zero [50]. It is expressed as follows:

$$RSS + \lambda \sum_{j=1}^{p} |\beta_j| \tag{3}$$

Elastic net regression [53] includes features of both $L_1$ and $L_2$ penalty terms and is expressed as follows:

$$RSS + \lambda \sum_{j=1}^{p} \beta_j^2 + \lambda \sum_{j=1}^{p} |\beta_j| \tag{4}$$

K-fold cross validation was used to verify predictive accuracy of the regularized models [49]. Root mean squared error (RMSE) was calculated to quantify the test error. The cross

validation procedure can be expressed as follows:

$$CV_k = \frac{1}{k}\sum_{i=1}^{k} RMSE \tag{5}$$

For each regularized model that was developed, the optimal model type (Ridge, LASSO, Elastic net) and optimal λ values were selected using a 10-fold cross validation that minimized the RMSE of predictions. Model types are represented by value $\alpha$, where $\alpha = 0$ represents Ridge, $\alpha = 1$ represents LASSO, and $0 < \alpha < 1$ represents Elastic net. The optimal λ, which represents regularization strength, determines the overall complexity of each model. R packages glmnet (v4.1.8) [54] and caret (v6.0.94) [55] and bestTune() were used to perform regularized regression and cross validation of models.

## Results

### Descriptive and statistical analysis results

Table 2 provides a description of the distribution of the study population characteristics. The study participants comprised primarily of female nursing students (85.3%) with a majority of

**Table 2. Study population characteristics.**

| Features | N (% of population) |
|---|---|
| **Biological sex** | |
| Female | 139 (85.3%) |
| Male | 23 (14.1%) |
| Prefer not to answer | 1 (0.6%) |
| **Age** | |
| 18–20 | 86 (52.8%) |
| 20–49 | 77 (47.2%) |
| **Race** | |
| White | 100 (61.3%) |
| Black or African American | 20 (12.3%) |
| Asian | 19 (11.7%) |
| Other | 24 (14.7%) |
| **Year of study** | |
| Freshman | 5 (3.1%) |
| Sophomore | 37 (22.7%) |
| Junior | 59 (36.2%) |
| Senior | 26 (16%) |
| Other | 36 (22%) |
| **Level of education** | |
| High school diploma/GED | 92 (56.4%) |
| Bachelors, Masters or other graduate degree | 71 (43.6%) |
| **Birth Place** | |
| Inside the United States | 147 (90%) |
| Outside the United States | 16 (10%) |
| **Home Language** | |
| Only English | 114 (70%) |
| Language other than English | 48 (29.4%) |
| Did not answer | 1 (0.6%) |

white race group (61.3%). Year of study was spread across multiple categories ranging from freshman to senior and other. Majority of students were born within the United States (90%) and spoke only English at home (70%). For our study, calculated Cronbach's alpha associated with the IPAQ is 0.66, with the MPAM-R is 0.95, with the EBBS is 0.82, and with the PASSS is 0.92.

To quantify the outcome of interest, PABS, for each participant, responses from each question in the EBBS related to barriers was summed. These responses are on a Likert scale, ranging from 1 to 4. Resulting PABS ranges from 14 to 49 with a median value of 28 and mean value of 26.5 and a standard deviation of 5.6 in the overall study population. Lower PABS indicates that an individual experiences less barriers, while a higher PABS indicates that the individual experiences more barriers.

Results show that PABS varies between different groups in study population. For example, the participants with a completed Bachelors/Masters or other graduate degree experience more barriers to engage in PA (PABS mean = 27.9, 95% Confidence Interval (CI): [24.89, 28.63]) than those with a high school diploma/GED (PABS mean = 25.3, 95% CI: [23.07, 25.95]). Furthermore, participants who are born in the US experience less barriers to PA (PABS mean = 26.1, 95% CI: [24.1, 26.4]) than those born outside of the US (PABS mean = 29.6, 95% CI: [23.1, 32.4]). Participants who only speak English at home experience less barriers to PA (PABS mean = 26.2, 95% CI: [24.5, 27.0]) than those who speak a language other than English at home (PABS mean = 27.1, 95% CI: [23.0, 27.8]).

Little's Test for missingness [42] was performed for variables in each validated survey tool and p-values were generated for each test under a 95% confidence interval. For our study, the p-value associated with IPAQ variables is 0.000510, with the MPAM-R is 0.992, with the EBBS is 0.228, with the PASSS is 0.772, with Health data and sharing variables (PhenX SDOH Toolkit) is 0.957, and with demographic variables (PhenX SDOH Toolkit) is 0. Variables belonging to the MPAM-R, EBBS, PASSS, and Health data and sharing tools have non-significant p-values, indicating that this data is MCAR.

Statistical tests were used for testing the differences between the categories of population characteristics as shown in Table 2. Normality assumption was tested via the Shapiro-Wilk test and homogeneity of variances assumption was tested via Levene's test. While homogeneity of variance was met, normality assumption was violated, warranting the use of non-parametric tests. The Mann-Whitney Test yielded a p-value of 0.113 for biological sex, 0.0045 for age category, 0.0044 for level of education, 0.0088 for participant birthplace, and 0.129 for home language. The Kruskal-Wallis test resulted in p-values of 0.055 for race and 0.093 for year of study. Higher age, higher education level, lower family income, birth place outside of U.S. and home language other than English were statistically significantly associated with higher PABS indicating individuals experiencing more barriers to engage in PA.

### Predictive analysis results

For each intersection of NIMHD domain-level of influence combination shown in Table 1, stepwise regression, regularized regression, and k-fold cross validation were performed. After model comparison and validation, four final models were selected to determine the optimal $\alpha$ and $\lambda$ values that minimized RMSE. Regression coefficients to be included in each final model were selected at the intersection of regression coefficient paths with the optimal $\lambda$ value (Fig 2). For each model, the outcome variable of interest, $y$, is defined as the individual-level PABS. Independent variables, given by $x_i$, are summarized in Table 3. Resulting final model parameters and performance metrics are summarized in Table 4.

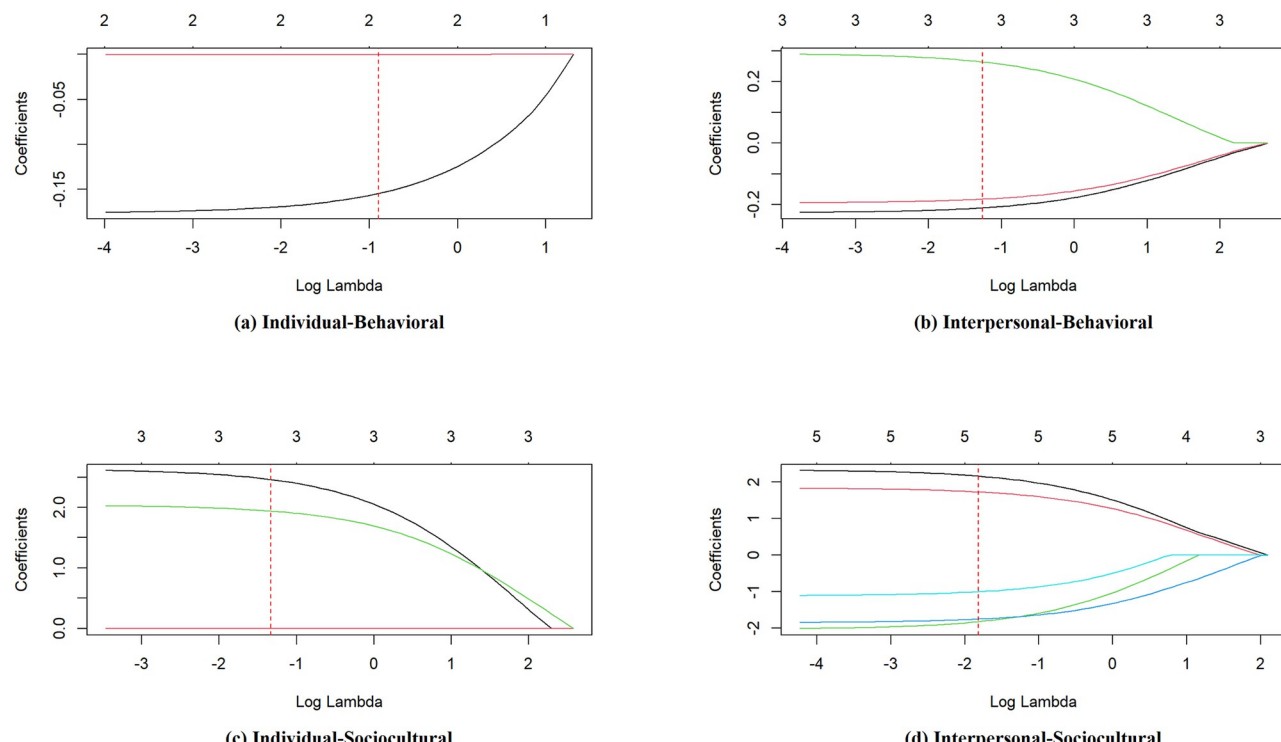

**Fig 2. Plots of regression coefficients vs. $log(\lambda)$ values for each domain-level of influence within the adapted NIMHD Framework.** As $log(\lambda)$ increases, regression coefficients are shrunk towards 0. Each model's regression coefficients are selected at the intersections with optimal $log(\lambda)$ values, shown by red dashed lines. (a) For Individual-Behavioral influences, two coefficients are selected for at $log(0.41)$. (b) For Interpersonal-Behavioral influences, three coefficients are selected for at $log(0.28)$. (c) For Individual-Sociocultural influences, three coefficients are selected for at $log(0.26)$. (d) For Interpersonal-Sociocultural influences, five coefficients are selected for at $log(0.16)$.

**Table 3. Independent variables used in final models.**

| Model | Independent Variable | Description | Range of Values |
|---|---|---|---|
| 1 | $x_1$ | Interest in PA as a motive | 7–49 |
| | $x_2$ | Total leisure-time PA (MET-minutes/week) | $\mathbb{R}_{\geq 0}$ |
| 2 | $x_3$ | Emotional support (e.g., advice, positive messages, encouragement) | 7–28 |
| | $x_4$ | Instrumental support (e.g., having appropriate gear, financial support) | 7–28 |
| | $x_5$ | Validation support (e.g., seeking out others for social comparison, relative status in a group) | 7–28 |
| 3 | $x_6$ | Total family income in the last year | $\mathbb{R}_{\geq 0}$ |
| | $x_7$ | Attained a bachelor's degree or higher | 0 or 1 |
| | $x_8$ | Male biological sex | 0 or 1 |
| 4 | $x_9$ | Does not make health decisions using a smartphone or tablet | 0 or 1 |
| | $x_{10}$ | Does not track health with a wearable device | 0 or 1 |
| | $x_{11}$ | Does not make health decisions with a wearable device | 0 or 1 |
| | $x_{12}$ | Does not text a health provider | 0 or 1 |
| | $x_{13}$ | Unsure of texting a health provider | 0 or 1 |

**Table 4. Regression statistics for final models.**

| Model | $\alpha$ | $\lambda$ | RMSE | $R^2$ | MAE | RMSESD | $R^2$ SD | MAESD |
|---|---|---|---|---|---|---|---|---|
| 1 | 0.55 | 0.41 | 5.07 | 0.22 | 4.10 | 1.06 | 0.17 | 0.77 |
| 2 | 0.10 | 0.28 | 5.07 | 0.21 | 4.00 | 1.09 | 0.20 | 0.73 |
| 3 | 0.10 | 0.26 | 5.36 | 0.18 | 4.28 | 0.85 | 0.18 | 0.63 |
| 4 | 0.10 | 0.16 | 5.45 | 0.16 | 4.39 | 0.83 | 0.15 | 0.61 |

$\alpha$ = model type, $\lambda$ = regularization strength, RMSE = Root Mean Square Error, $R^2$ = Coefficient of determination, MAE = Mean Absolute Error, RMSESD = Root Mean Square Error Standard Deviation, $R^2$ SD = Coefficient of determination Standard Deviation, MAESD = Mean Absolute Error Standard Deviation

**Model 1: Individual-behavioral domain-level of influence intersection.**  Model 1 is an Elastic Net with $\alpha = 0.55$ and $\lambda = 0.41$ (Fig 2A). Two independent variables were retained in the final model as shown in Eq (6).

$$y = 32.07 - 0.15(x_1) - 3.00 \cdot 10^{-4}(x_2) \tag{6}$$

Variable $x_1$ represents interest as a motivator for PA and takes on values from 7 to 49, with higher values indicating that interest is a greater motivator for PA. Variable $x_2$ represents total leisure-time PA in MET-minutes/week, and takes on values in $\mathbb{R}_{\geq 0}$. The intercept term is 32.07. Interpreting the coefficients, $x_1$ is negatively associated with outcome variable PABS: for every unit increase in interest as a motivator, PABS decreases by 0.15. Similarly, $x_2$ is negatively associated with PABS, and for every unit increase in leisure-time PA, PABS decreases by $3.00 \cdot 10^{-4}$. Variable $x_1$ is notably more influential than $x_2$, where its coefficient $\beta_1$ is approximately 3 orders of magnitude higher than $\beta_2$. Model 1 explained 22% of the variance in the PABS outcome ($R^2 = 0.22$) and had an RMSE value of 5.07.

**Model 2: Interpersonal-behavioral domain-level of influence intersection.**  Model 2 is an Elastic Net with $\alpha = 0.10$ and $\lambda = 0.28$ (Fig 2B). The final model retains three independent variables, all representing different forms of social support as shown in Eq (7).

$$y = 29.17 - 0.21(x_3) - 0.18(x_4) + 0.26(x_5) \tag{7}$$

The independent variables with higher values indicate that an individual experiences more social support. The intercept of the model is 29.17. Variables $x_3$ and $x_4$ are negatively associated with PABS, where experiencing one unit higher emotional support (e.g., advice, positive messages, encouragement) and instrumental social support (e.g., having appropriate gear, financial support) is associated with PABS decreasing by 0.21 and 0.18, respectively. Interestingly, $x_5$ is positively associated with the PABS outcome, increasing by 0.26 for every unit greater validation support (e.g., seeking out others for social comparison, relative status in a group). This suggests that receiving more validation support increases the real or perceived barriers an individual experiences. Model 2 explained 21% of the variance in the PABS outcome ($R^2 = 0.21$) and had an RMSE of 5.07.

**Model 3: Individual-sociocultural domain-level of influence intersection.**  Model 3 is an Elastic Net with $\alpha = 0.10$ and $\lambda = 0.26$ (Fig 2C). The final model retains 3 independent variables as shown in Eq (8).

$$y = 26.10 - 5.75 \cdot 10^{-6}(x_6) + 1.94(x_7) + 2.46(x_8) \tag{8}$$

Variable $x_6$ represents income and takes on values in $\mathbb{R}_{\geq 0}$. Variable $x_7$ represents having attained a bachelor's degree or higher and $x_8$ represents having a male biological sex. Both $x_7$

and $x_8$ take on binary values of 0 or 1. The intercept is given as 26.10. For every \$10000 greater in family income, PABS was 0.0575 less. In contrast, if $x_7$ and $x_8$ take on values of 1, PABS increases by 1.94 and 2.46, respectively. These results suggest that having higher levels of education and being of male biological sex are associated with higher barriers to PA, while having a higher income is associated with lower barriers. Model 3 explained 18% of the variance in the PABS outcome ($R^2 = 0.18$) and had an RMSE of 5.36.

**Model 4: Interpersonal-sociocultural domain-level of influence intersection.** Model 4 is an Elastic Net with $\alpha = 0.10$ and $\lambda = 0.16$ (Fig 2D). The final model retains 5 independent variables as shown in Eq (9).

$$y = 26.83 + 2.16(x_9) + 1.73(x_{10}) - 1.82(x_{11}) - 1.75(x_{12}) - 1.00(x_{13}) \tag{9}$$

These variables measure the use of technology as a means to monitor health, and take on binary values of 1 if the statement is true, and 0 otherwise. For example, $x_9$ takes the value 1 if the statement "Does not make health decisions using a smartphone or tablet" is true for participant, 0 otherwise. Variables $x_9$ ("Does not make health decisions using a smartphone or tablet") and $x_{10}$ ("Does not track health with a wearable device") are positively associated with barriers to PA, and greater PABS by 2.16 and 1.73, respectively, for every unit increase. On the other hand, reporting yes to $x_{11}$ ("Does not make health decisions with a wearable device"), $x_{12}$ ("Does not text a health provider"), and $x_{13}$ ("Unsure of texting a health provider") were associated with lower barriers to PA. Model 4 explained 16% of the variance in the PABS outcome ($R^2 = 0.16$) and had an RMSE of 5.45.

Comparing all models, Model 1 contained the least number of predictors, the highest alpha value, and highest lambda value. This indicates that, compared to other models, there was a more aggressive shrinkage of regression coefficients to 0. Conversely, Models 2, 3, and 4 all had the same alpha value that leaned towards $L_2$ regularization, which promoted the inclusion of relevant variables. Model 1 generated the smallest prediction error and had the best predictive accuracy of all models. The reason for developing for separate regression models for each domain-level intersection was the low predictive performance of a model that tried to predict PABS associated with all four NIMHD domain-influence levels at once.

## Discussion

In this study, we focused on answering two research questions. To explore the first question ("How can we quantify self-reported barriers to engage in PA using a scoring system?"), we utilized two domains (behavioral and sociocultural environment) and two levels of influence (individual and interpersonal) from the NIMHD framework and matched validated survey questionnaires to these domain-influence intersections. To explore the second study question ("How can we use individual, interpersonal, behavioral and socio-cultural factors to predict this score in nursing student population?"), we developed regularized regression models to predict PABS for each domain-influence intersection.

To the best of our knowledge, this is the first study that utilized the NIMHD framework and regularized regression models to study multi-level factors hindering PA. Previous studies have implemented ordinary least squares (OLS) regression and various logistic regression models to explore the intersection between health disparities and PA, but regularized regression methods have not been explored extensively [13, 21, 56, 57]. When using linear regression, collinearity between parameters and parameters canceling each other, among other reasons, can impact model performance. Our models overcome these challenges by constraining the coefficient parameters by incorporating a penalty term to achieve models with better predictive power.

Key findings from Model 1 show the importance of interest in PA as an intrinsic motivation factor to reduce barriers to PA. Additionally, engaging in leisure time PA is associated with lower barriers score. This is as expected, as adults with lower barriers to PA are more likely to active during their leisure time [3].

Model 2 highlights the importance of social support in reducing barriers to PA with interesting findings. More emotional support (e.g., advice, encouragement) and instrumental support (e.g., having appropriate gear, financial support) are associated with lower PABS. This is in line with previous studies demonstrated that social and environmental supports are key to increasing and maintaining PA [3]. Conversely, individuals reporting higher validation support (e.g., seeking social comparison or holding a relative status in a group) also indicated higher PABS. This unexpected correlation might find explanation in the Social Determination Theory, wherein seeking validation support acts as an external motivator [58]. Consequently, those seeking such support might lack intrinsic motivation, often perceiving more barriers than benefits to PA. In contrast, internally motivated individuals, who tend to focus on the benefits and enjoyment PA, perceive fewer barriers. Our study supports the development of PA interventions focusing on enhancing intrinsic motivation, as proposed in the Self-determination Theory, by addressing facilitating factors.

Model 3 focuses on highest level of education and annual family income. Results show that higher family income is associated with lower PABS, as expected. Interestingly, higher education is associated with higher PABS. This could be related to age being a confounder. PA tends to decline from young to middle age adulthood. This in part, may be due to the life events and transitions (e.g., shifting family and work responsibilities) that may be barriers to being physically active during midlife [59].

Model 4 indicates the importance of decision making and tracking functionalities of health technology that helps reduce the barriers to PA. Wearable technologies and complimentary smartphone and web applications can address barriers and promote PA behavior change techniques such as self monitoring, goal setting, and data sharing for social support [60]. Wearable activity trackers can also encourage lifestyle PA spread throughout the day and reduce the barriers associated with structured forms of PA that often require more time and resources [61].

## Practical contributions

There is a growing need for developing health-informatics driven solutions that target future healthcare workforce to identify populations who are at risk of low PA levels. Previous studies have shown the usefulness of influencing PA with health technology, such as smart devices or wearables [62]. Our regression models identified the importance of decision making and tracking functionalities of health technology to reduce the barriers to PA. Wearable technologies and complimentary smartphone and web applications can address barriers and promote PA behavior change techniques such as self-monitoring, goal setting and data sharing for social support [60]. Wearable activity trackers can also encourage lifestyle PA spread throughout the day and reduce barriers associated with structured forms of PA that often require more time and resources [61]. Further, utilizing the NIMHD Research Framework as proposed in this study can inform development of interventions to address disparities in experiencing barriers to PA. Understanding and acknowledging the barriers to PA that stem from socioeconomic factors, like educational background and income accessibility, should be integral in shaping and executing future interventions and policy frameworks. In addition, nurses serve as role models for patients' health behaviors. Nurses who experience less barriers to engage in PA are better equipped to encourage patients to healthy PA behaviors, promoting a wellness culture in the healthcare system [11, 63].

## Limitations and future directions

Our study has several limitations. First, we used survey data from nursing student population at a single academic institution, which may have introduced sampling bias. The conclusions derived from this limited sample may not be broadly applicable and have limited generalizability. To address this, we recommended increasing the sample size and expanding the study to multiple settings in future studies.

Our study has utilized self-reported survey-based data, which is aligned with the PA literature. Studies that investigate PA levels have utilized self-reported questionnaires or devices (e.g., accelerometers, pedometers) [64]. Of these, self-reported questionnaires are common due to their ease of access and cost-effectiveness [65], however, are prone to recall bias. Barriers to PA have been similarly measured with various questionnaires [37, 66, 67]. Our models focused on the individual and interpersonal levels of influence and the behavioral and socio-cultural domains.

Future studies can expand the selection of validated survey tools and utilize more domains and levels of influence within the NIMHD Framework to increase the included variables in predictive models. In addition, future studies can incorporate data from wearable devices. This is particularly important to improve the model performance as all predictive models in our study had low adjusted $R^2$ values which suggests that there may be other factors impacting barriers to PA that our models have not considered. These potentially missed variables could be captured with wearable device data. Low adjusted $R^2$ values could also indicate that linear models are not capturing the complex relationships between multi-level factors associated with barriers to PA. Future studies should consider non-linear models.

Our study highlights that there are significant limitations in the published literature related to populations, particularly in patient-facing healthcare professions like nursing, who may experience higher barriers to engage in PA. This needs to be addressed in future studies by exploring a variety of individual, social and environmental factors specifically targeting the nursing, and future nursing, workforce populations. In conclusion, intrinsic motivation, social and emotional support, and use of health technologies for tracking and decision making purposes were the significant predictors associated with barriers to PA. Developing and employing health informatics solutions to support the monitoring, data sharing, and communication that elicit social support and motivation with complementary health technologies may be critical tactics to prevent barriers and support healthy, active living among the nursing profession.

## Supporting information

**S1 File. Final version of REDCap survey.**
(PDF)

**S1 Table. Missing value percentages for independent variables.**
(PDF)

**S1 Data. Participant-level dataset used in descriptive and predictive analyses.**
(XLSX)

## Acknowledgments

The authors thank all participants for their participation and nursing faculty at the The Elaine Marieb College of Nursing for dedicating class time for the study data collection.

## Author Contributions

**Conceptualization:** Muge Capan, Amanda Paluch, Joohyun Chung.

**Data curation:** Muge Capan, Lily Bigelow, Yukti Kathuria.

**Formal analysis:** Lily Bigelow, Yukti Kathuria.

**Funding acquisition:** Muge Capan.

**Investigation:** Lily Bigelow, Amanda Paluch.

**Methodology:** Muge Capan.

**Project administration:** Muge Capan.

**Resources:** Amanda Paluch, Joohyun Chung.

**Software:** Lily Bigelow, Yukti Kathuria.

**Supervision:** Muge Capan.

**Validation:** Muge Capan.

**Writing – original draft:** Muge Capan, Lily Bigelow, Yukti Kathuria.

**Writing – review & editing:** Amanda Paluch, Joohyun Chung.

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
