## [Decision Letter · Decision Letter 0]

1 Apr 2024

PONE-D-24-02933Analysis of Multi-level Barriers to Physical Activity Among Nursing Students Using Regularized RegressionPLOS ONE

Dear Dr. Capan,

Thank you for submitting your manuscript to PLOS ONE. After careful consideration, we feel that it has merit but does not fully meet PLOS ONE’s publication criteria as it currently stands. Therefore, we invite you to submit a revised version of the manuscript that addresses the points raised during the review process.

**It is a great pleasure to read your manuscript. Your discussion on healthcare providers’ well-being addresses a topic of utmost importance. However, as noted in the reviewers’ comments, some revisions are necessary before a final decision on publication can be made.**

**Editor’s note: I recommend that the authors merge their introduction and literature review to focus more on their targeted questions: the identification of barriers and the association of these barriers with physical activity (PA). The manuscript contains redundancies that could be condensed for clarity. Additionally, minor changes could enhance the manuscript's quality. For instance, results discussed in the methods section, such as Cronbach’s alpha results, should be moved to the results section. Moreover, methodological details discussed in the discussion section, such as those found in Line 408-412, would be better suited for the methods section. Lastly, common statistical statements, such as 'p<0.05 considered statistically significant,' could be avoided. I suggest the authors delve into the clinical/practical significance of their approaches and explain why such approaches are valuable.**

We look forward to receiving your revised manuscript.

Kind regards,

Hao Wang

Academic Editor

PLOS ONE

Journal Requirements:

University of Massachusetts Amherst -Elaine Marieb Center for Nursing and Engineering Innovation Faculty Pilot Grant supporting the study entitled "Exploratory Study to Identify Multi-level Factors of Physical Health and Well-being in Nursing Profession" (06/2023 – 06/2024, PI: M. Capan).

The authors thank all participants for their participation and nursing faculty at the The Elaine Marieb College of Nursing for dedicating class time for the study data collection. The authors would like to acknowledge the Elaine Marieb Center for Nursing and Engineering Innovation at the University of Massachusetts Amherst for their financial support through the internal pilot grant entitled: ”Exploratory Study to Identify

Multi-level Factors of Physical Health and Well-being in Nursing Profession”

University of Massachusetts Amherst -Elaine Marieb Center for Nursing and Engineering Innovation Faculty Pilot Grant supporting the study entitled "Exploratory Study to Identify Multi-level Factors of Physical Health and Well-being in Nursing Profession" (06/2023 – 06/2024, PI: M. Capan).

4. In the online submission form, you indicated that Data cannot be shared publicly because of personal information to protect data privacy. De-identified data are available from the corresponding author at University of Massachusetts Mechanical and Industrial Engineering Department upon request.

Additional Editor Comments:

It is a great pleasure to read your manuscript. Your discussion on healthcare providers’ well-being addresses a topic of utmost importance. However, as noted in the reviewers’ comments, some revisions are necessary before a final decision on publication can be made.

Editor’s note: I recommend that the authors merge their introduction and literature review to focus more on their targeted questions: the identification of barriers and the association of these barriers with physical activity (PA). The manuscript contains redundancies that could be condensed for clarity. Additionally, minor changes could enhance the manuscript's quality. For instance, results discussed in the methods section, such as Cronbach’s alpha results, should be moved to the results section. Moreover, methodological details discussed in the discussion section, such as those found in Line 408-412, would be better suited for the methods section. Lastly, common statistical statements, such as 'p<0.05 considered statistically significant,' could be avoided. I suggest the authors delve into the clinical/practical significance of their approaches and explain why such approaches are valuable.

Reviewers' comments:

Reviewer's Responses to Questions

**Comments to the Author**

1. Is the manuscript technically sound, and do the data support the conclusions?

Reviewer #1: Yes

Reviewer #2: Yes

2. Has the statistical analysis been performed appropriately and rigorously? 

Reviewer #1: Yes

Reviewer #2: I Don't Know

3. Have the authors made all data underlying the findings in their manuscript fully available?

Reviewer #1: Yes

Reviewer #2: Yes

4. Is the manuscript presented in an intelligible fashion and written in standard English?

Reviewer #1: Yes

Reviewer #2: Yes

5. Review Comments to the Author

**Reviewer #1:** Dear author,

I found the topic of your manuscript to be interesting and well thought out. The background, Method, Results, and Discussion sections are good written. After reviewing it, I wanted to share two points on Method with you:

You mentioned that one of the inclusion criteria for participants was consent to participate in study. This special criterion is an exclusion criterion.

You reported about 14% data were missed. Although you Following imputation for retaining data, missing data might influence the result. As you applied electronic questionnaire, you could have coped with data missed randomly. It seems that missing data were nonrandom. That means students avoided to response to some questions. Please describe this limitation in the “Discussion”.

Discussion: Please update the references used in your manuscript.

**Reviewer #2: **Thank you for the opportunity to review this interesting study “Analysis of Multi-level Barriers to Physical Activity Among Nursing Students Using Regularized Regression”. This is a cross-sectional study that used survey data to quantify self-reported barriers to physical activity and used multiple factors to predict physical activity barrier scores (PABS). This study attempts to use the National Institute on Minority Health and Health Disparities (NIMHD) Research Framework to identify factors that may contribute to a participant’s overall physical activity barrier score. Participants were Nursing students currently enrolled in a nursing degree program at the study institution. The total number of participants was 163. The participants completed a 174-question survey that combined elements from multiple surveys to model the multiple domains of the NIMHD framework when assessing physical activity barrier score. The authors then performed statistical analysis to develop four different linear regression models to predict PABS.

This is an interesting study that is well-written with some issues that affect its quality.

Abstract

- Overall well-written but exceeds the 300 word limit

Introduction/Relevant Literature

- Minor grammatical changes: “Improving nurses’ PA relies on better understanding the factors acting as barriers to engage in PA specifically in nursing population”. “However, there is a lack of research that studies PA, and particularly barriers to PA disproportionately experience by certain populations…”

Materials and Methods

- Table 1: elements of the table are listed above and below the table. Please only list the elements of the table underneath and include PhenX Social Determinants of Health (SDOH) toolkit

Results

- 296-305: This section compares several means and reports associated confidence intervals. Consider including associated p-values for these data.

-

- Why does fig 2. Only includes data for level of education. Why was level of education chosen for this example?

- 366-376: Variable x8 in this model is “attained a master’s or other graduate degree” and is reported as significantly influential in this model with a coefficient of 5.64. This model has only 7 participants in the “masters or other graduate degree” category compared to 92 and 64 for high school diploma/GED and bachelors degree respectively. Please address this limitation.

- 371-372: Variable x6 is family income and is has the coefficient − 5.00 · 10−6

o It might be useful to give an example of how this part of the formula works. Does it mean a decrease in PABS of 5 for every 1 million dollars in family income?

Discussion

- Paragraph lines 398-405 is redundant and can be removed.

6. PLOS authors have the option to publish the peer review history of their article (what does this mean?). If published, this will include your full peer review and any attached files.

Reviewer #1: No

Reviewer #2: No

---

## [Author Response · Author response to Decision Letter 0]

5 May 2024

Please see submission file "Response to Reviewers". The content is copy pasted below as well for your reference.

PONE-D-24-02933

Analysis of Multi-level Barriers to Physical Activity Among Nursing Students Using Regularized Regression

Editor’s note

I recommend that the authors merge their introduction and literature review to focus more on their targeted questions: the identification of barriers and the association of these barriers with physical activity (PA). The manuscript contains redundancies that could be condensed for clarity. 

Thank you for your valuable feedback. In the revised manuscript we merged introduction and literature review sections to improve the content and flow of the manuscript and reduced repetitive information.

Additionally, minor changes could enhance the manuscript's quality. For instance, results discussed in the methods section, such as Cronbach’s alpha results, should be moved to the results section. 

We moved the Cronbach’s alpha results to the results section, as recommended.

Moreover, methodological details discussed in the discussion section, such as those found in Line 408-412, would be better suited for the methods section. 

We absolutely agree. We moved the discussion of methodological details to methods section.

Lastly, common statistical statements, such as 'p<0.05 considered statistically significant,' could be avoided. 

We made sure that common statistical statements were removed.

I suggest the authors delve into the clinical/practical significance of their approaches and explain why such approaches are valuable.

Thank you for this insight. We believe that the revised manuscript provides a clear discussion of the clinical and practical implications and significance, and how studies like our can lay the foundation for important future research directions.

Review Comments to the Author

Reviewer #1: 

I found the topic of your manuscript to be interesting and well thought out. The background, Method, Results, and Discussion sections are good written. 

Thank you for your feedback, we greatly appreciate it.

After reviewing it, I wanted to share two points on Method with you: You mentioned that one of the inclusion criteria for participants was consent to participate in study. This special criterion is an exclusion criterion.

We revised the language in Study Population section to clarify “consent to participate” as an exclusion criterion.

You reported about 14% data were missed. Although you Following imputation for retaining data, missing data might influence the result. As you applied electronic questionnaire, you could have coped with data missed randomly. It seems that missing data were nonrandom. That means students avoided to response to some questions. Please describe this limitation in the “Discussion”.

Thank you for bringing up this important issue regarding missingness in the data. We agree that survey-based data, as utilized in this study, may exhibit missingness that can be nonrandom as participants (in this case nursing students) may have intentionally refused answering some questions. We added this limitation in the discussion section for clarification and as potential future analysis area.

Discussion: Please update the references used in your manuscript.

We reviewed and updated the refences.

Reviewer #2: Thank you for the opportunity to review this interesting study “Analysis of Multi-level Barriers to Physical Activity Among Nursing Students Using Regularized Regression”. This is a cross-sectional study that used survey data to quantify self-reported barriers to physical activity and used multiple factors to predict physical activity barrier scores (PABS). This study attempts to use the National Institute on Minority Health and Health Disparities (NIMHD) Research Framework to identify factors that may contribute to a participant’s overall physical activity barrier score. Participants were Nursing students currently enrolled in a nursing degree program at the study institution. The total number of participants was 163. The participants completed a 174-question survey that combined elements from multiple surveys to model the multiple domains of the NIMHD framework when assessing physical activity barrier score. The authors then performed statistical analysis to develop four different linear regression models to predict PABS. This is an interesting study that is well-written with some issues that affect its quality.

Thank you for your feedback, we greatly appreciate it.

Abstract

- Overall well-written but exceeds the 300 word limit

The abstract in the revised manuscript is under the 300 word limit.

Introduction/Relevant Literature

- Minor grammatical changes: “Improving nurses’ PA relies on better understanding the factors acting as barriers to engage in PA specifically in nursing population”. “However, there is a lack of research that studies PA, and particularly barriers to PA disproportionately experience by certain populations…”

We appreciate the reviewer pointing out the minor grammatical changes needed. We revised the manuscript to ensure the grammatical errors are eliminated. 

Materials and Methods

- Table 1: elements of the table are listed above and below the table. Please only list the elements of the table underneath and include PhenX Social Determinants of Health (SDOH) toolkit

Results

We listed the elements under the table as recommended. 

- 296-305: This section compares several means and reports associated confidence intervals. Consider including associated p-values for these data.

We have added the p-values as recommended. 

- Why does fig 2. Only includes data for level of education. Why was level of education chosen for this example?

This Figure aimed to provide one example of descriptive data visualization applied while exploring the data. We agree that it may cause confusion to share the boxplot og one particular variable used in the models discussed later in the manuscript. We removed this Figure and revised the section.

- 366-376: Variable x8 in this model is “attained a master’s or other graduate degree” and is reported as significantly influential in this model with a coefficient of 5.64. This model has only 7 participants in the “masters or other graduate degree” category compared to 92 and 64 for high school diploma/GED and bachelors degree respectively. Please address this limitation.

We absolutely agree with reviwer’s point. We combined categories. Level of education now had two categories: High school diploma/GED 92 (56.4%) and Bachelors, Masters or other graduate degree 71 (43.6%) as shown in revised Table 2. All four models were re-run, all results updated accordingly. In revised modeling results, Model 3 contains variable x6 (“total family income in the last year”), variable x7 (“attained a bachelor's degree or higher”), and a new variable x8 (“male biological sex”). Variable x6 has a coefficient of 5.75 · 10−6, variable x7 has a coefficient of 1.94, and variable x8 has a coefficient of 2.46. These results show that education is still an influential predictor, but that male biological sex is also influential for this population. After combining the categories for education, there were slight improvements in the predictive performance Model 3. The changes resulting from this have been reflected in Table 2, Table 4, Figure 3, and in Model 3.

- 371-372: Variable x6 is family income and is has the coefficient − 5.00 · 10−6

o It might be useful to give an example of how this part of the formula works. Does it mean a decrease in PABS of 5 for every 1 million dollars in family income?

We have provided an updated explanation for this variable in the revised manuscript. Every $10000 increase in family income is associated with a PABS decrease by 0.0575. 

Discussion

- Paragraph lines 398-405 is redundant and can be removed.

Thank you, we removed these redundant elements.

---

## [Editor Report · Decision Letter 1]

9 May 2024

Analysis of Multi-level Barriers to Physical Activity Among Nursing Students Using Regularized Regression

PONE-D-24-02933R1

Dear Dr. Capan,

We’re pleased to inform you that your manuscript has been judged scientifically suitable for publication and will be formally accepted for publication once it meets all outstanding technical requirements.

Kind regards,

Hao Wang

Academic Editor

PLOS ONE

Additional Editor Comments (optional):

Authors responded appropriately to all the comments. The revised manuscript is now considered suitable for publication.
---

## [Editor Report · Acceptance letter]

14 May 2024

PONE-D-24-02933R1 

PLOS ONE

Dear Dr. Capan, 

I'm pleased to inform you that your manuscript has been deemed suitable for publication in PLOS ONE. Congratulations! Your manuscript is now being handed over to our production team.

Kind regards, 

on behalf of

Dr. Hao Wang 

Academic Editor

PLOS ONE